# High-Temperature Reaction Mechanism of Molybdenum Metal in Direct Coal Liquefaction Residue

Chunling Wu [1,2], Linge Ma [3], Yufei Zhu [3], Xuqiang Guo [2], Yongli Wu [1], Zhen Wu [1], Xian Zhang [1] and Lihua Hou [1,*]

1　School of Chemical Engineering, Ordos Institute of Technology, Ordos 017000, China
2　State Key Laboratory of Heavy Oil Processing, China University of Petroleum-Beijing, Changping District, Beijing 102249, China
3　National Institute of Clean-and-Low-Carbon Energy, Changping District, Beijing 102211, China
*　Correspondence: lh.hou@163.com; Tel.: +86-138-4867-3967

**Abstract:** In this paper, the extraction residue of direct coal liquefaction residue-$DCLR_{(ER)}$ was used as raw material. The high-temperature reaction mechanism of Mo compound in $DCLR_{(ER)}$ was investigated using a synchronous thermal analyzer and the Factsage database. The high temperature reaction of $DCLR_{(ER)}$-$MoO_3$ in an oxygen atmosphere consists of pyrolysis of organic components at 400–600 °C, molybdenum trioxide sublimation at 747–1200 °C, and a stable stage at 600–747 °C. The thermal reaction process of the $DCLR_{(ER)}$-$MoS_2$ system in the oxygen atmosphere involves the pyrolysis of unreacted coal and asphaltene, the oxidation of molybdenum sulfide at 349–606/666 °C, the diffusion of $MoO_3$ at 606/666–85 °C, and the sublimation reaction process of $MoO_3$ at 854–1200 °C. The results show that the lower heating rate can promote the oxidation of the Mo compound and the sublimation of molybdenum trioxide. On the other hand, the oxides of aluminum, calcium, and iron in $DCLR_{(ER)}$ can inhibit the oxidative pyrolysis efficiency of the $DCLR_{(ER)}$-$MoS_2$ system.

**Keywords:** direct coal liquefaction residue; molybdenum; high temperature; reaction mechanism





## 1. Introduction

Under the reaction conditions of high temperature, high pressure, and catalyst, the coal and the hydrogen supplied by the hydrogen-supplying solvent undergo a series of complex reactions in the reactor, such as hydrocracking, polycondensation, and the removal of heterosis atoms [1–3]. Then, the structure of coal, and especially the carbon–hydrogen ratio, will undergo significant changes, converting solid coal into liquid oil. The technological process is defined as direct coal liquefaction [4,5]. As an important by-product of the direct coal liquefaction system, the yield of the direct coal liquefaction residue can reach about 30% of the raw coal. In coal liquefaction engineering, the majority of the coal is hydrogenated in the reactor, resulting in the generation of liquid oil, a trace of gas, and water [6]. In the coal liquefaction project, the coal is hydrogenated in the reactor, most of which produces liquid oil with a wide distribution of fractions, a small amount of gas and water, etc. The vacuum distillation process separates oils with heavier fractions and all solid materials, including catalysts, coal ash (inorganic minerals, etc.), and unreacted coal from the feedstock coal slurry into solid–liquid [7]. Oils with heavier solvent components and distillation ranges are distilled from the vacuum column line and overhead, while the heavy oils and bituminous substances (asphaltene and preasphaltene) and all solids are distilled from the bottom of the vacuum column discharge [8]. Direct coal liquefaction residue (DCLR) is the primary by-product of coal liquefaction after cooling and solidification [9]. The yield of coal liquefaction residue is about 30% of raw coal (wt.), while the inorganic minerals (minerals) account for about 20% (wt.) of the total residue. Inorganic minerals include minerals from the coal liquefaction itself and newly formed and residual metal catalysts during the liquefaction process [10]. The molybdenum catalyst and molybdenum iron composite catalyst [11,12]

have been developed based on the nano-FeOOH catalyst in industrial application [13]. It is found that $MoS_2$, $MoO_3$, and iron-molybdenum composite catalysts can improve the coal conversion in the direct coal liquefaction process. The iron–molybdenum composite catalyst developed by Shenhua Shanghai Research Institute can increase the coal conversion of Shendong Coal by 2.8–3.1% [14]. To achieve a high conversion rate without increasing the cost of the catalyst used, the direct coal liquefaction technology should be investigated for the physical and chemical properties of the highly reactive metal components in the direct coal liquefaction residue, and more effective recycling technologies employing reduced highly reactive metals should be developed. This is a cost-effective way to increase the efficiency and energy efficiency of direct coal liquefaction, as well as to realize the commercial value of the process and maintain competitiveness.

Metal molybdenum recovery from waste catalyst mainly includes the wet method, the fire method, the electrolysis method, the ultrasonic/microwave-assisted method, the direct conversion method, and other recovery methods [15]. The metal in the spent catalyst structure usually exists in the form of metal sulfide. Before the leaching process, Roasting Pretreatment converts metal sulfides into metal oxides, which are then converted into soluble metal salts through a dissolution reaction, so that the metal oxides contained in the solid waste catalyst are dissolved and transferred to the solution, as shown in Formula (1) [16].

$$MoO_3 + 2NaHSO_4 = MoO_2SO_4 + Na_2SO_4 + H_2O. \tag{1}$$

The wet recovery process, as a traditional process technology, has the advantages of stable technology and strong operability. It is suitable for the recovery of molybdenum in different molybdenum-containing media. The disadvantage is that a large number of strong acids and bases will be used in the wet process, which requires high corrosion resistance of equipment, and the subsequent acid-base treatment process is involved, so the process flow is relatively long.

Carbon has a strong ability to reduce rare metals in waste catalysts at high temperature, and plasma arc has the characteristics of high temperature. Zhaopeng Zhu [17] used the method of plasma furnace heating to recover molybdenum, nickel, and other metals. They fused the alumina carrier to form slag to separate metals and residue. The valuable metals of the alloy smelted and enriched with waste catalyst account for about 70%. However, the incomplete process hinders the further promotion of the process.

Other emerging methods, such as ion exchange [18], physical adsorption [19], extraction, and other technologies [20,21], have good results when recovering molybdenum components from wastewater. However, they have high selectivity for molybdenum-containing material media. For the recovery of molybdenum in solid materials, the combination of pretreatment processes such as dissolution is also required. Combined with the complex particularity of the components of coal direct liquefaction residue, we carried out research on the technology of recovering molybdenum catalyst using a high-temperature oxidation–low-temperature coagulation method [22].

In order to improve the recovery method, it is necessary to study the reaction mechanism of $DCLR_{(ER)}$-$MoO_3$ and $DCLR_{(ER)}$-$MoS_2$ systems under high-temperature conditions to recover molybdenum from the catalyst in the $DCLR_{(ER)}$ based on high-temperature reaction. The organic components in DCLR can be separated via organic solvent extraction, to obtain extraction residue of direct coal liquefaction residue ($DCLR_{(ER)}$). $DCLR_{(ER)}$ was used as raw material and mixed with $MoO_3$ and $MoS_2$ powder at a certain mass ratio to prepare experimental samples. The high-temperature reaction mechanism of Mo compound in $DCLR_{(ER)}$ was investigated using a synchronous thermal analyzer and the Factsage database.

## 2. Results and Discussion

### 2.1. MoO₃ Sublimation Mechanism

Sublimation is the reaction step in which atoms in the lattice break away from neighboring atoms and enter the gas phase [23]. It is possible to control the rate of vaporization by appropriately adjusting the sublimation conditions to achieve the goal of increasing or decreasing its rate through an investigation of these steps. The evaporation and condensation of molybdenum trioxide are often accompanied by the formation of various polymers, such as dimer ($Mo_2O_6$), trimer ($Mo_3O_9$), and tetramer ($Mo_4O_{12}$). According to current studies [24,25], when the $MoO_3$-$O_2$ system was operated at a pressure of 0.1 MPa and a temperature less than 600 °C, molybdenum was in a condensed state. The system changed to the gas phase as the temperature increased, and the formation of the polymer was observed in the gas phase. Each Mo atom forms Mo=O with two oxygen atoms and a Mo–O double bond with two oxygen atoms in the trimer ($Mo_3O_9$). A ring structure is formed by three oxygen atoms and three molybdenum atoms. The structure tends to form a stable tetrahedral structure, so it is further transformed into a tetramer ($Mo_4O_{12}$). The process is as follows:

$$MoO_3(s) \rightarrow Mo_3O_9(g) \rightarrow Mo_4O_{12}(g). \tag{2}$$

The pyrolysis characteristics of molybdenum trioxide in the $MoO_3$-$O_2$ system are shown in Figure 1. Commercial molybdenum trioxide powder was used as the raw material; it can be seen from the figure that its microscopic morphology is spherical particles. Commercial $MoO_3$ was placed in an oxygen flow of 15 mL/min while being heated at a rate of 20 °C/min to observe the weight loss. Figure 1 depicts the TG-DSC analysis curve of molybdenum trioxide, which exhibits two distinct endothermic peaks at 722.4 °C and 787.5 °C, respectively. The TG curve corresponding to the endothermic peak at 722.4 °C exhibited no weight loss, indicating that molybdenum trioxide has not yet sublimated at this temperature, which should correspond to the heat change caused by molybdenum trioxide melting. The TG value corresponded to the endothermic peak at 787.5 °C demonstrated significant weight loss, with a weight loss rate of 96.0% between 741.5 °C and 1200 °C, which was primarily due to molybdenum trioxide sublimation. The microscopic morphology of the sublimation–crystallization product is strip-shaped (Figure 2).

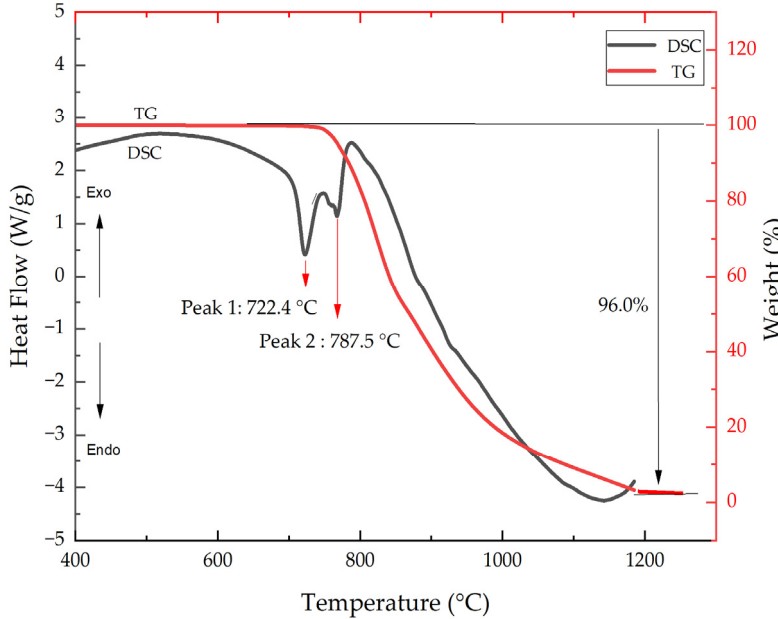

**Figure 1.** TG/DSC curve of molybdenum trioxide pyrolysis.

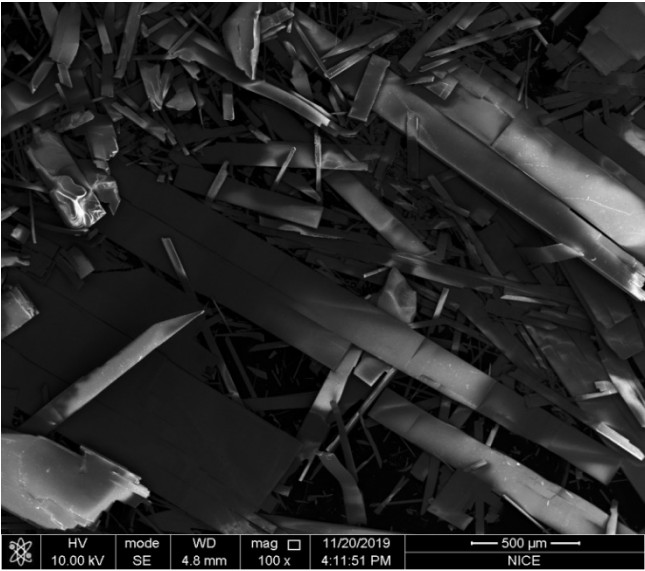

**Figure 2.** SEM image of the sublimation–crystallization product.

*2.2. Mechanism of MoS$_2$ Oxidation*

The oxidation of molybdenum sulfide is a complex reaction system involving gas–solid and solid–solid reactions at the same time. The main reactions are as follows:

$$MoS_2 + O_2 = MoO_3 + 2SO_2. \tag{3}$$

The equilibrium constant of the above reaction can be calculated by Formulas (4) and (5):

$$lgK_T = -\frac{\Delta G}{2.303RT} \tag{4}$$

$$\Delta G/(J \cdot mol^{-1}) = -1074744.08 + 61.38TlgT - 2.26 \times 10^{-2}T^2 + 2.60 \times 10^{-6}T^3 + 57.74T \tag{5}$$

At 873 K, the equilibrium constant of the reaction reaches 1052, indicating that the oxidation of molybdenum sulfide in the air or pure oxygen is irreversible. Marin [26] proposed a kinetic model and hypothesized that MoO$_2$ was an important intermediate product in the oxidation of molybdenum sulfide to molybdenum trioxide, and the process consisted of two steps:

$$MoS_2 + 3O_2(g) = MoO_2 + 2SO_2(g) \tag{6}$$

$$MoO_2 + 1/2O_2(g) = MoO_3 \tag{7}$$

The activation energies of the two steps were similar, with values of 23.0 kJ/mol and 20.8 kJ/mol, respectively. It can be considered that the diffusion of oxygen through the oxide porous product layer controlled the reaction rates for the two steps. There was no formation of a product layer during the initial stage of the reaction, and the conversion of molybdenum sulfide and oxygen to molybdenum dioxide proceeded rapidly. The rate of the reaction decreased gradually as the MoO$_3$ product layer formed, and the reaction changed from initial chemical reaction control to diffusion control.

The pyrolysis characteristics of molybdenum sulfide in the MoS$_2$-O$_2$ system are shown in Figure 2. Commercial molybdenum sulfide powder was used as the raw material; it can be seen from the figure that its microscopic morphology is in the form of irregular flakes. It was placed in an oxygen flow of 15 mL/min while being heated at a rate of 20 °C/min to observe the weight loss.

A clear endothermic peak was observed at 456.3 °C on the TG-DSC analysis curve of molybdenum sulfide, as depicted in Figure 3. Molybdenum sulfide undergoes oxidation with a weight loss rate of 13.8% in the range of 200–527.5 °C at TG. Furthermore, the DSC curve demonstrated a weak endothermic peak at 777.3 °C, and significant weight loss was observed at TGA. The weight loss rate reached 82.2% in the temperature range of 737.6 °C to 1200 °C, which could be attributed to molybdenum trioxide sublimation. The microscopic morphology of the sublimation crystallization product is a strip product of different sizes (Figure 4).

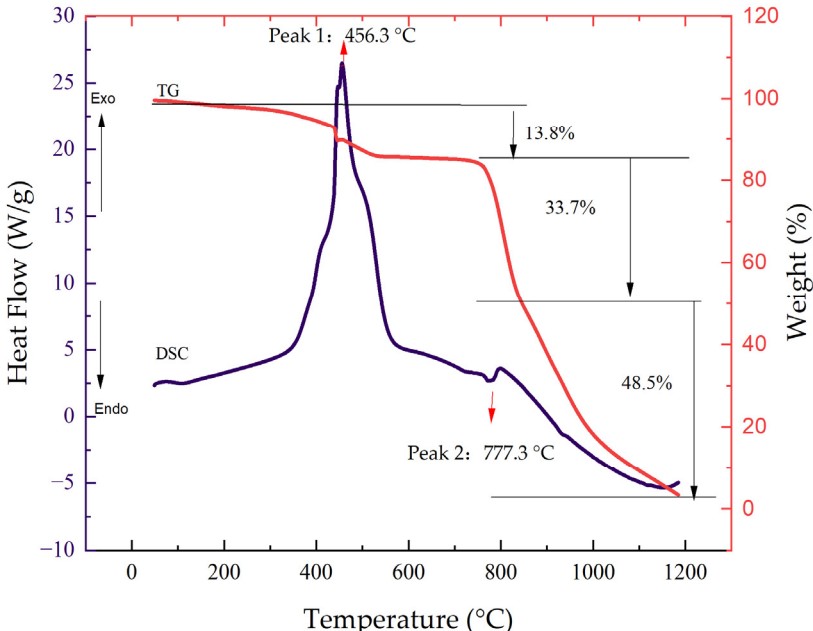

**Figure 3.** TG/DSC curve of molybdenum sulfide pyrolysis.

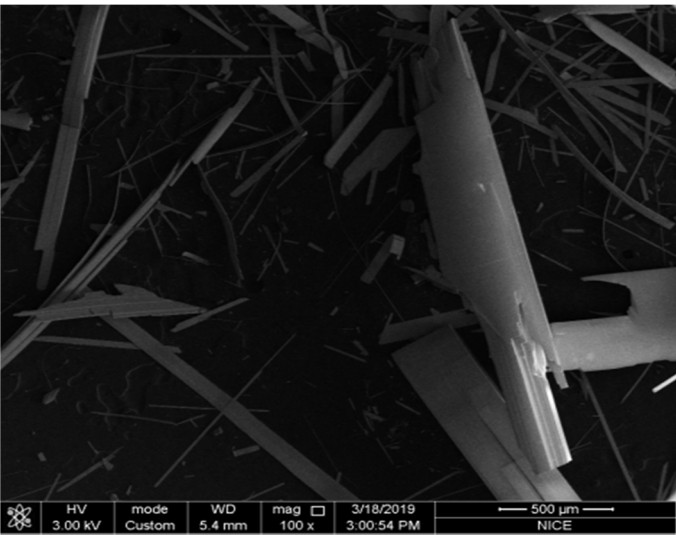

**Figure 4.** SEM image of the sublimation crystallization product.

### 2.3. High-Temperature Reaction Mechanism of DCLR$_{(ER)}$-MoO$_3$ System

According to the analysis in Section 2.2, the sublimation of molybdenum trioxide is mainly transformed into a gas phase through the transformation of solid molybdenum trioxide through the polymer, and the corresponding process involves the diffusion of the gas–solid phase. When molybdenum trioxide is present in the DCLR extraction system, its

diffusion is limited by the surrounding solid layer. Figure 5 illustrates the DSC curves for samples with various $DCLR_{(ER)}$-$MoO_3$ mass ratios (30:70, 50:50, and 70:30).

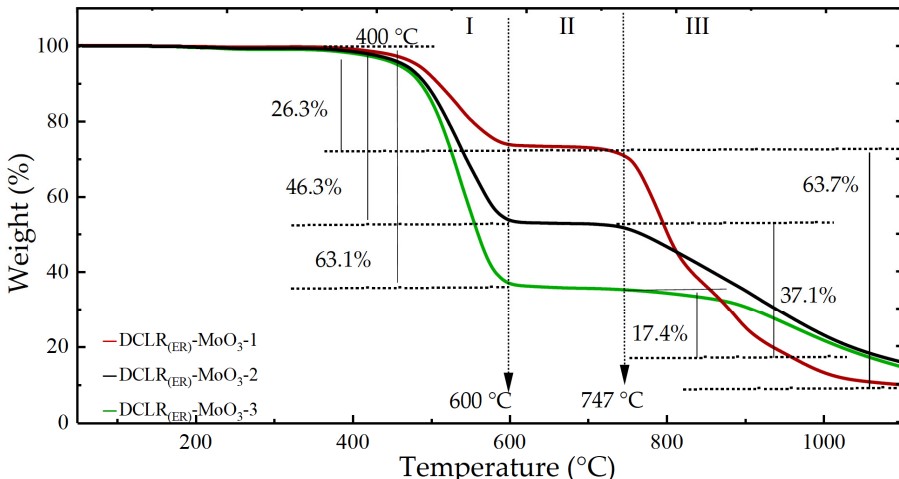

**Figure 5.** TG curve of $DCLR_{(ER)}$-$MoO_3$ Pyrolysis.

As illustrated in Figure 5, samples with varying mass ratios exhibited similar thermogravimetric characteristics that can be roughly classified into three stages: the first stage (400~600) °C, the second stage (600~747) °C, and the third stage (747~1200) °C $DCLR_{(ER)}$-$MoO_3$-3 had the highest weight-loss rate in the first stage, at 63.1%. At the same stage, the weight-loss rates for $DCLR_{(ER)}$-$MoO_3$-2 and $DCLR_{(ER)}$-$MoO_3$-1 were 46.3% and 26.3%, respectively, corresponding to the characteristics of $DCLR_{(ER)}$-$MoO_3$-1 < $DCLR_{(ER)}$-$MoO_3$-2 < $DCLR_{(ER)}$-$MoO_3$-3. Based on the component content of the samples, it can be considered that the first stage is primarily the pyrolysis reaction process of organic components in DCLR, such as coal and asphalt. The third stage curve followed the opposite trend to the first stage. $DCLR_{(ER)}$-$MoO_3$-1, $DCLR_{(ER)}$-$MoO_3$-2, and $DCLR_{(ER)}$-$MoO_3$-3 showed weight-loss rates of 60.8%, 36.8%, and 20.4% in the third stage, and the sublimation rates of $MoO_3$ were 86.8%, 73.6%, and 68.0% in the third stage, respectively. The diffusion of oxygen and molybdenum trioxide was inhibited as the mass ratio of $DCLR_{(ER)}$ increased, and the sublimation rate of molybdenum trioxide decreased gradually, $DCLR_{(ER)}$-$MoO_3$-1 > $DCLR_{(ER)}$-$MoO_3$-2 > $DCLR_{(ER)}$-$MoO_3$-3. It was revealed, based on the component content of the experimental samples, as well as the pyrolysis characteristics of molybdenum trioxide discussed in Section 2.1, that the sublimation reaction process of $MoO_3$ occurred primarily during this interval. When temperatures were increased from 600 °C to 747 °C in the second stage, no obvious weight loss was observed, and the weight loss of all three samples was less than 3%, indicating that there was no characteristic reaction. The results are shown in Table 1.

**Table 1.** Results of $DCLR_{(ER)}$-$MoO_3$ thermal analysis with different mass ratios.

| Reaction Stage | Reaction Process | Weight Loss/% | | |
|---|---|---|---|---|
| | | $DCLR_{(ER)}$-$MoO_3$-1 | $DCLR_{(ER)}$-$MoO_3$-2 | $DCLR_{(ER)}$-$MoO_3$-3 |
| I (400~600) °C | Pyrolysis of unreacted coal and asphaltene in $DCLR_{(ER)}$ | 26.3 ± 0.4 | 46.3 ± 0.4 | 63.1 ± 0.4 |
| II (600~747) °C | Stable stage | 2.8 ± 0.4 | 2.1 ± 0.4 | 1.8 ± 0.4 |
| III (747~1200) °C | $MoO_3$ sublimation reaction | 60.8 ± 0.4 | 36.8 ± 0.4 | 20.4 ± 0.4 |
| I + II + III (400~1200) °C | $DCLR_{(ER)}$ pyrolysis-$MoO_3$ sublimation | 89.9 ± 0.4 | 85.2 ± 0.4 | 85.3 ± 0.4 |

### 2.4. High-Temperature Reaction Mechanism of DCLR$_{(ER)}$-MoS$_2$ System

DCLR$_{(ER)}$-MoS$_2$-3 was used as the experimental sample to investigate the reaction mechanism of DCLR at high temperatures in the presence of a catalyst in the form of molybdenum sulfide, and its pyrolysis characteristics were observed by heating up to 1200 °C at rates of 10 °C/min, 15 °C/min, and 30 °C/min, respectively. The variable temperature TG analysis of DCLR$_{(ER)}$-MoS$_2$-3 and SEM of the sublimation crystallization product is shown in Figures 6 and 7.

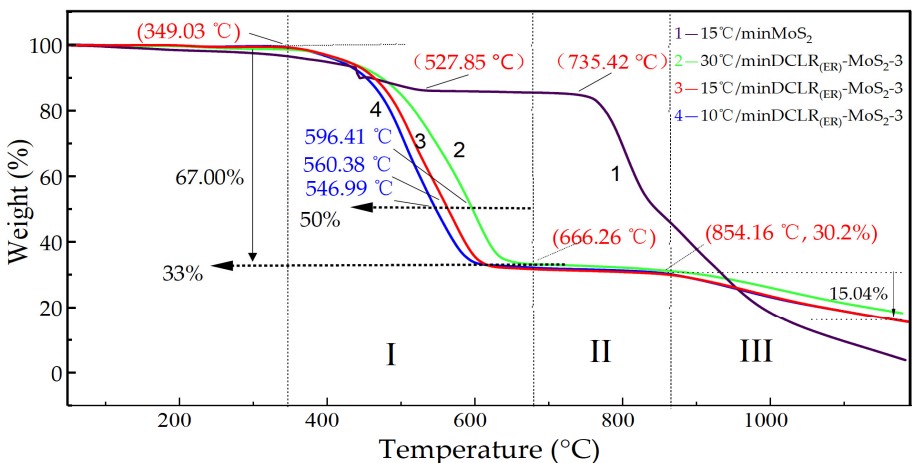

**Figure 6.** TG curves of DCLR$_{(ER)}$-MoS$_2$ pyrolysis.

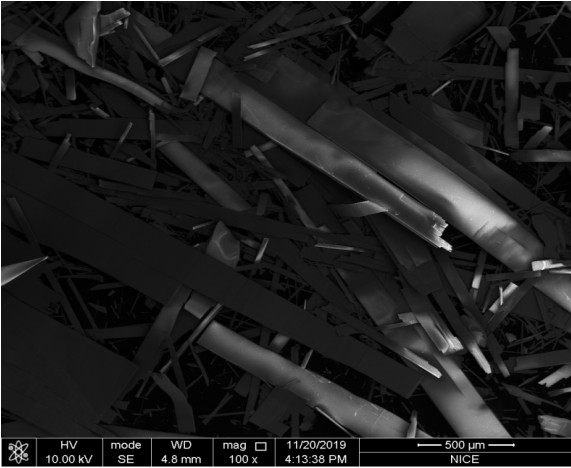

**Figure 7.** SEM image of the sublimation crystallization product.

The pyrolysis characteristics of DCLR$_{(ER)}$-MoS$_2$-3 samples were significantly different from those of pure molybdenum disulfide, as shown in Figure 6, even though their curves varied at different heating rates. Pure MoS$_2$ samples began to lose weight gradually from the start of heating up to 527 °C, at which point they lost 14% of their weight, whereas DCLR$_{(ER)}$-MoS$_2$-3 began to react at 349 °C, with significant weight loss. This is the initial reaction stage. The first stage reached 606 °C at a heating rate of 10 °C/min and 15 °C/min, while the final stage reached 666 °C at a heating rate of 30 °C/min. As can be seen, various heating rates affect the reaction during the first stage. Temperatures of 10 °C/min, 15 °C/min, and 30 °C/min at 50% weight loss were 547 °C, 560 °C, and 596 °C, respectively, with the characteristics of 10 °C/min < 15 °C/min < 30 °C/min. The first stage temperatures of 10 °C/min and 15 °C/min overlapped at 606 °C. It was determined that the reaction at this stage was primarily the pyrolysis of DCLR$_{(ER)}$ and the oxidation of MoS$_2$ based on the reaction mechanism of DCLR$_{(ER)}$-MoS$_2$ and pure MoS$_2$. Unlike the first stage curve,

TG curves with different heating rates almost overlapped until the reaction temperature reached 854 °C in the second stage. The weight loss in this region was less than 3.5%, indicating a stable stage primarily involving the diffusion of the product molybdenum trioxide. After 854 °C, the rate of thermal weight loss increased, reaching 15.04% between 854 °C and 1200 °C. When compared to the $MoO_3$ and $DCLR_{(ER)}$-$MoO_3$ high-temperature reaction rules in Sections 2.2 and 2.3, it is clear that the sublimation reaction of $MoO_3$, the oxidation product, takes place primarily in this region. In comparison to pure $MoO_3$ and $DCLR_{(ER)}$-$MoO_3$, the sublimation onset temperature of oxidation product $MoO_3$ in $DCLR_{(ER)}$-$MoS_2$ increased by 113 °C and 107 °C, respectively. In the third stage, there was no significant difference in the weight-loss curves of different heating rates, as shown in Figure 4, and the weight-loss curves of 10 °C/min and 15 °C/min heating rates almost overlapped. The weight-loss rate was significantly greater at the two heating rates than at 30 °C/min at the same temperature. The slower heating rate increased the diffusion of molybdenum trioxide in the first stage, which is favorable for molybdenum trioxide sublimation. Therefore, 15 °C/min was chosen as the best reaction heating rate in this experiment, taking reaction time into account, to achieve a higher product yield. The results are shown in Table 2.

**Table 2.** Results of $DCLR_{(ER)}$-$MoS_2$ thermal analysis at different heating rates.

| Reaction Stage | Reaction Process | Weight Loss/% | | |
|---|---|---|---|---|
| | | 30 °C/min | 15 °C/min | 10 °C/min |
| I (349~606/666) °C | Pyrolysis of unreacted coal and asphaltene in $DCLR_{(ER)}$; molybdenum sulfide oxidation | $67.0 \pm 0.4$ | $67.0 \pm 0.4$ | $67.0 \pm 0.4$ |
| II (606/666~854) °C | $MoO_3$ diffusion | $2.8 \pm 0.4$ | $3.3 \pm 0.4$ | $3.3 \pm 0.4$ |
| III (854~1200) °C | $MoO_3$ sublimation reaction | $12.4 \pm 0.4$ | $15.04 \pm 0.4$ | $15.2 \pm 0.4$ |
| I + II + III (349~1200) °C | $DCLR_{(ER)}$ pyrolysis-$MoO_3$ sublimation | $82.2 \pm 0.4$ | $85.3 \pm 0.4$ | $85.5 \pm 0.4$ |

The weight loss rate in the third stage of $DCLR_{(ER)}$-$MoS_2$-3 and $DCLR_{(ER)}$-$MoO_3$-3 was 17.4% and 15.0%, respectively, according to the above analysis. When the weight-loss rate of $MoO_3$ is taken into consideration, it can be seen that 60% of the molybdenum oxide in this system is converted into the gas phase. $DCLR_{(ER)}$ has been shown to inhibit the oxidation of $MoS_2$ and the sublimation of $MoO_3$. On one hand, it is caused by the effect of $DCLR_{(ER)}$ on $MoO_3$ diffusion. However, the sublimation of $MoO_3$ is reduced as a result of the reaction of other metal oxide impurities in $DCLR_{(ER)}$-$MoO_3$ under such conditions to form a by-product layer that is difficult to pyrolyze. The high-temperature reaction process of the $DCLR_{(ER)}$-$MoS_2$ system is shown in Figure 8.

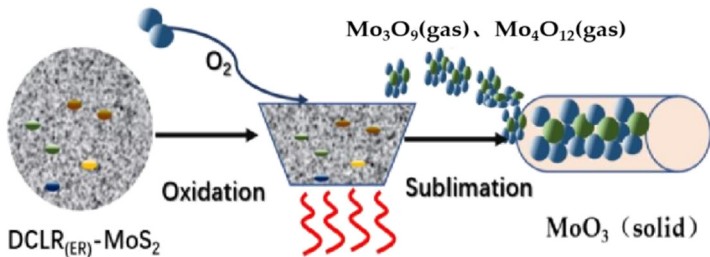

**Figure 8.** The illustration of high-temperature reaction process of $DCLR_{(ER)}$-$MoS_2$ system.

In this experiment, X-fluorescence spectrum analysis was performed on $DCLR_{(ER)}$ and residue after sublimation reaction with high-temperature oxidation. It can be seen that the metal components of the raw coal in the reaction system mainly include silicon, aluminum, calcium, iron, and other metals, containing a small amount of magnesium,

potassium, sodium, titanium, zinc, manganese, strontium metal, and accompanied by sulfur, phosphorus, and other elements. The composition of the ash of raw coal is shown in Table 3. These metal components react with some $MoO_3$ to form a by-product layer that is difficult to pyrolyze, preventing a small amount of $MoO_3$ from sublimation.

**Table 3.** Composition of the ash of raw coal.

| Composition | Content/% | Composition | Content/% |
|---|---|---|---|
| $SiO_2$ | $49.74 \pm 0.03$ | $K_2O$ | $1.78 \pm 0.02$ |
| $Al_2O_3$ | $17.33 \pm 0.03$ | $TiO_2$ | $0.83 \pm 0.04$ |
| CaO | $17.4 \pm 0.02$ | MnO | $0.2 \pm 0.02$ |
| $Fe_2O_3$ | $6.3 \pm 0.03$ | SrO | $0.16 \pm 0.01$ |
| MgO | $1.78 \pm 0.04$ | $P_2O_5$ | $0.09 \pm 0.01$ |
| $Na_2O$ | $1.88 \pm 0.02$ | $SO_3$ | $2.19 \pm 0.01$ |

Figure 9 shows the phase distribution calculated at equilibrium by thermodynamic software FactSage 7.3 using FactPS, FToxid, and aiMP databases [27–29]. $Fe_2(SO_4)_3$ was the predominant metal that formed between 400 and 700 °C, and iron oxide and calcium oxide began reacting with molybdenum trioxide at 710 °C to form $Fe_2Mo_3O_{12}$ and $CaMoO_4$, respectively. The reaction between metal impurities and molybdenum trioxide in $DCLR_{(ER)}$ above has an inhibitory effect on the high-temperature reaction catalysis of molybdenum recovery from $DCLR_{(ER)}$, thus affecting the recovery rate.

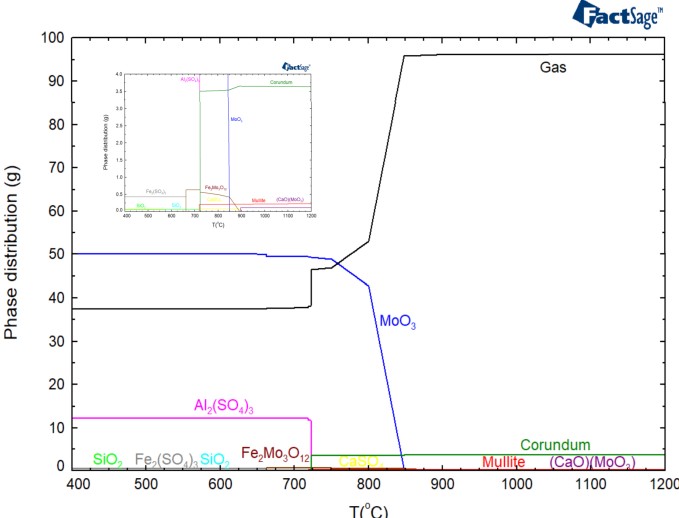

**Figure 9.** Thermodynamic calculation results of $DCLR_{(ER)}$ high-temperature reaction.

The above analysis shows that controlling the temperature range of the reaction system can be used to avoid or reduce the influence of metal components such as iron and calcium on the sublimation diffusion reaction of molybdenum trioxide.

### 2.5. The Transformation of Molybdenum Catalyst in DCLR$_{(ER)}$

X-ray diffraction analysis results of the product are shown in Figure 10. It can be seen that the diffraction peaks of the product are consistent with the standard spectrum of molybdenum trioxide PDF 05-0508. The diffraction peaks at 12.75°, 23.30°, 25.65°, 27.30°, and 38.95° coincide with the 020, 110, 040, 021, and 060 crystal planes of orthorhombic molybdenum trioxide, and the product belongs to orthorhombic molybdenum trioxide. The peak value of the product increased significantly at 020, 040 and 060, with a high degree of anisotropy. After two phase transformations, the crystal structure of molybdenum trioxide changed significantly.

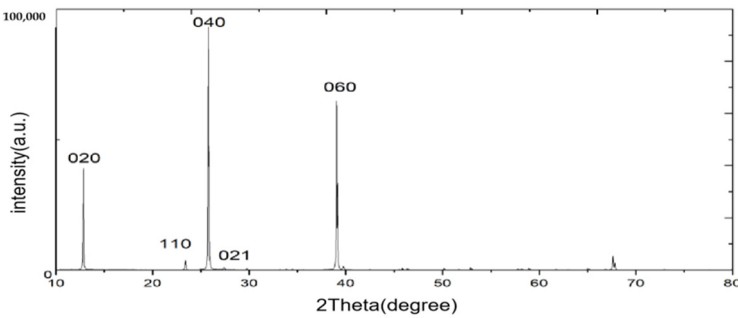

**Figure 10.** XRD spectrogram of the products from DCLR$_{(ER)}$-MoS$_2$.

In summary, the residual molybdenum catalyst in the direct coal liquefaction residue mainly exists in the form of molybdenum sulfide. Under the condition of high temperature and oxygen, the molybdenum sulfide is oxidized to form molybdenum trioxide, which is further sublimated into gas phase. When the temperature of recovery section decreases, gaseous molybdenum trioxide will form molybdenum trioxide crystal [22]. The main reactions are as follows:

$$MoS_2(s) + 3O_2 \xrightarrow{(349 \sim 666) \ ^\circ C} MoO_2(s) + 2SO_2 \tag{8}$$

$$MoO_2(s) + 1/2O_2 \xrightarrow{(349 \sim 666) \ ^\circ C} MoO_3(s) \tag{9}$$

$$MoO_3(s) \xrightarrow{(666 \sim 1200) \ ^\circ C} Mo_3O_9(g) \xrightarrow{(666 \sim 1200) \ ^\circ C} Mo_4O_{12}(g) \xrightarrow{(400 \sim 125) \ ^\circ C} MoO_3(s) \tag{10}$$

## 3. Materials and Methods

### 3.1. Experimental Material

The information and properties of experimental materials and reagents are listed in Table 4.

**Table 4.** Materials and reagents.

| Reagent | Chemical Formula | Purity | Melting Point/°C | Boiling Point/°C |
|---|---|---|---|---|
| Nitrogen | N$_2$ | 99.99% | −210 | −196 |
| Oxygen | O$_2$ | 99.99% | −218 | −183 |
| Molybdenum disulfide | MoS$_2$ | A.R. | 2375 | 450 |
| Molybdenum trioxide | MoO$_3$ | A.R. | 795 | 1150 |

The raw material DCLR$_{(ER)}$ is the solid obtained from Shenhua coal direct liquefaction residue (DCLR) after solvent extraction. Its appearance is shown in Figure 11.

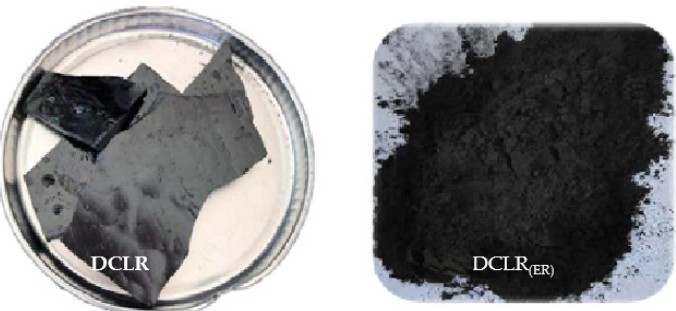

**Figure 11.** The images of DCLR and DCLR$_{(ER)}$.

The reagents of molybdenum disulfide and molybdenum oxide were purchased from Aladdin®Reagent (Shanghai, China). Their appearance is shown in Figure 12. Nitrogen and oxygen were purchased from Dalian Date Gas Co., Ltd (Dalian, China).

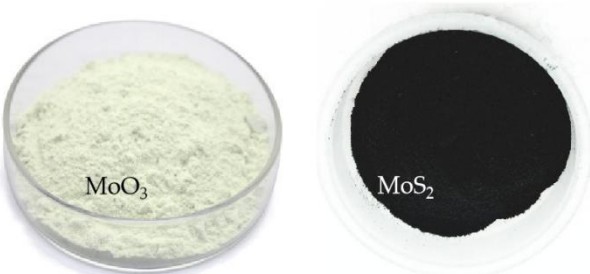

**Figure 12.** The images of molybdenum oxide and molybdenum disulfide.

*3.2. Experimental Equipment*

The information of the equipment used in this experiment is shown in Table 5.

**Table 5.** Equipment list.

| Instrument Name | Instrument Model | Manufacturer | City, Country |
| --- | --- | --- | --- |
| Circulating water vacuum pump | SHZ-D | Yuhua Instrument Co. | Gongyi, China |
| Electric blast drying oven | KSD | KangYi electronic instrument factory | Guangzhou, China |
| Constant temperature shaker | THZ-98C | Shanghai Yiheng Co. | Shanghai, China |
| X-ray fluorescence spectrum analyzer | Axios 4400/40 | PANalytical B.V. | Almelo, The Netherlands |
| Synchronous thermal analyzer | TA SDT650 | American TA instrument Co. | New Castle, DE, USA |
| scanning electron microscope | FEI Nova Nano SEM 450 | Thermo Fisher Scientific Co. | Waltham, MA, USA |
| X-ray powder diffractometer | D8 Advance | Bruker AXS | Saarbrücken, Germany |
| Fourier infrared spectrometer | IRPrestige-21 | SHIMADZU Co. | Kyoto, Japan |

The high-temperature pyrolysis characteristics of the $DCLR_{(ER)}$ system were investigated in this experiment using a TA Discovery SDT 650 synchronous thermal analyzer. The experimental equipment was composed of a heating unit, a weight-testing unit, a system control unit, and a recording unit, as shown in Figure 13. Among them, the balance had a sensitivity of 0.1 mg, a calorimetric accuracy of 2%, a heating rate of 0.1~25 °C/min, and a temperature range of room temperature to 1500 °C.

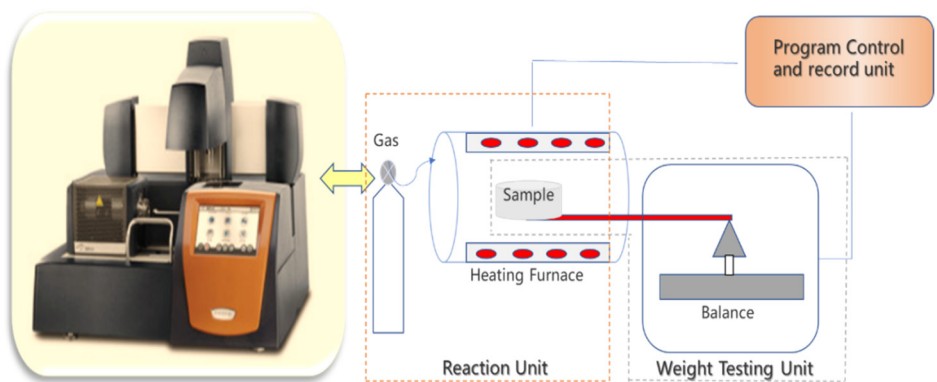

**Figure 13.** Synchronous thermal analyzer (TA Discovery SDT 650).

*3.3. Experiment Steps*

(1) Separation of organic matter and inorganic matter in DCLR. The DCLR is mixed with 1-butyl-3-methylimidazole nitrate at the mass ratio of 1:10. The mixture is put into a constant temperature shaking table at the extraction temperature of 40 °C and vibrated for 0.5 h. Then, the mixture is taken out and filtered. The extracted residue after filtration

is washed with deionized water and transferred to a vacuum oven to dry the solvent and water. The filtrate is separated from ionic liquid and organic matter via stripping with ionic water and filtration. The extraction residue of direct coal liquefaction residue ($DCLR_{(ER)}$) and organic matter are obtained, respectively, and the appearance is shown in Figure 14. The composition analysis results of $DCLR_{(ER)}$ are shown in Table 6.

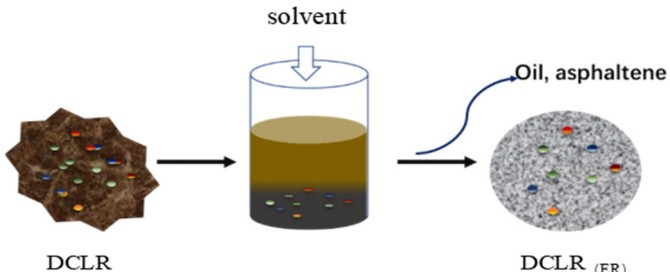

**Figure 14.** $DCLR_{(ER)}$ and organic matter separated by 1-butyl-3-methylimidazole nitrate.

**Table 6.** The analysis of the $DCLR_{(ER)}$ component.

| Industrial Analytical Components/% | Elemental Analysis Component/% | | | |
|---|---|---|---|---|
| $A_d$ | $C_d$ | $H_d$ | $N_d$ | $S_d$ |
| $42.3 \pm 0.15$ | $81.03 \pm 0.10$ | $4.26 \pm 0.11$ | $1.03 \pm 0.16$ | $3.62 \pm 0.10$ |
| Composition/% | | | | |
| $Al_2O_3$ | $SiO_2$ | $Fe_2O_3$ | CaO | $MoO_3$ | $SO_3$ |
| $7.38 \pm 0.03$ | $20.28 \pm 0.03$ | $28.66 \pm 0.03$ | $22.76 \pm 0.02$ | — | $14.01 \pm 0.01$ |

Notes: $A_d$ stands for ash content on a dry basis; $C_d$, $H_d$, $N_d$, $S_d$, $O_d$ stand for dry-basis carbon, hydrogen, nitrogen, sulfur and oxygen elements, respectively.

(2) Preparation of experimental samples. The experimental raw material $DCLR_{(ER)}$ and the molybdenum disulfide were mixed according to the mass ratios displayed in Table 7, transferred into a mortar, and ground into uniform particles.

**Table 7.** Sample information.

| | $DCLR_{(ER)}$-Fe/g | $MoS_2$/g | $MoO_3$/g |
|---|---|---|---|
| $MoS_2$ | 0 | 100 | 0 |
| $DCLR_{(ER)}$-$MoS_2$-1 | 30 | 70 | 0 |
| $DCLR_{(ER)}$-$MoS_2$-2 | 50 | 50 | 0 |
| $DCLR_{(ER)}$-$MoS_2$-3 | 70 | 30 | 0 |
| $MoO_3$ | 0 | 0 | 100 |
| $DCLR_{(ER)}$-$MoO_3$-1 | 30 | 0 | 70 |
| $DCLR_{(ER)}$-$MoO_3$-2 | 50 | 0 | 50 |
| $DCLR_{(ER)}$-$MoO_3$-3 | 70 | 0 | 30 |
| $DCLR_{(ER)}$ | 100 | 0 | 0 |

(3) We opened the nitrogen inlet valve and set the pressure to 0.1 MP before turning on the instrument to allow it to warm up. We removed the sample crucible and added 5–10 mg analytically pure $MoS_2$, analytically pure $MoO_3$, $DCLR_{(ER)}$-$MoO_3$-1, $DCLR_{(ER)}$-$MoO_3$-2, $DCLR_{(ER)}$-$MoO_3$-3, $DCLR_{(ER)}$-$MoS_2$-1, $DCLR_{(ER)}$-$MoS_2$-2, and $DCLR_{(ER)}$-$MoS_2$-3. Then, we inserted the crucible into the balance's sample end and set the oxygen gas flow to 15 mL/min. The heating rates were 10 °C/min, 15 °C/min, 30 °C/min, and gradually increased from 25 °C to 1200 °C. The weight and heat flow of samples were observed. The TG-DSC analysis results were used to investigate the high-temperature reaction mechanism of $DCLR_{(ER)}$-$MoO_3$ and $DCLR_{(ER)}$-$MoS_2$ systems.

(4) Based on the results of X-fluorescence analysis of $DCLR_{(ER)}$-$MoO_3$-2 components, the phase change and reaction mechanism of the reaction system were calculated using FactPS, FToxid, and AIMP databases in thermodynamic software FactSage 7.3.

## 4. Conclusions

The high-temperature reaction mechanism of the molybdenum metal in the oxygen atmosphere, as well as the $DCLR_{(ER)}$, was investigated using thermal analysis and calculation.

(1) The high-temperature reaction mechanism and law of direct coal liquefaction residue of containing molybdenum were revealed through investigation of the reaction characteristics of $MoO_3$-$O_2$, $MoS_2$-$O_2$, $DCLR_{(ER)}$-$MoO_3$-$O_2$, and $DCLR_{(ER)}$-$MoS_2$-$O_2$. The results show that the high-temperature reaction goes through three steps in Molybdenum metal components in coal direct liquefaction residues. The first stage is the pyrolysis reaction and $MoS_2$ oxidation reaction; the second stage is diffusion of $MoO_3$ and $MoS_2$ oxidation reaction; the third stage is the sublimation reaction process of $MoO_3$.

(2) Molybdenum trioxide was converted from solid to gaseous state in a continuous heating system in an oxygen atmosphere via the formation and transformation of a gaseous trimer and tetramer, which involved the melting and sublimation of molybdenum trioxide. Unlike the pure molybdenum trioxide, the high-temperature reaction of $MoO_3$ in $DCLR_{(ER)}$ can be divided into three stages. It consists of pyrolysis of organic components in $DCLR_{(ER)}$ at 400–600 °C, molybdenum trioxide sublimation at 747–1200 °C, and a stable stage at 600–747 °C. The content of $DCLR_{(ER)}$ in the reaction affects the overall efficiency and yield of molybdenum trioxide. In comparison to 70% and 50% $DCLR_{(ER)}$, the total weight loss rate of 30% $DCLR_{(ER)}$ increased by 4.6% and 4.7%, respectively. Meanwhile, the sublimation rates of molybdenum trioxide increased by 13% and 18.6%, respectively.

(3) The thermal reaction process of the $DCLR_{(ER)}$-$MoS_2$ system is divided into three stages. The first stage involves the pyrolysis of unreacted coal and asphaltene, as well as the oxidation of molybdenum sulfide at 349–606/666 °C; the second stage involves the diffusion of $MoO_3$, which results in weight loss in the range of 606/666–854 °C; and the third stage involves the sublimation reaction of $MoO_3$ at 854–1200 °C. The effect of heating rate on the reaction was studied. The lower heating rate enhanced the yield of the product by promoting the oxidation reaction and sublimation of molybdenum trioxide. The oxides of aluminum, calcium, and iron in $DCLR_{(ER)}$ can inhibit the oxidative pyrolysis efficiency of the $DCLR_{(ER)}$-$MoS_2$ system.

(4) Through this study, we have mastered the reaction mechanism of molybdenum metal catalyst in the residue of direct coal liquefaction under high-temperature and aerobic conditions. Compared with other spent catalyst systems, it is a complex system containing a variety of metals and organic matter. The results of this study can provide a theoretical basis for the development and improvement of our subsequent high-temperature sublimation and low-temperature condensation process used to recover the residual molybdenum catalyst in the residue of direct coal liquefaction.

**Author Contributions:** C.W., X.G. conceptualization, writing—review and editing, and supervision. C.W., Y.Z. methodology, the main part of the experimental and data curation, writing—original draft preparation. L.H., L.M., Y.W., Z.W. and X.Z. partial experimental and data collection. All authors have read and agreed to the published version of the manuscript.

**Funding:** This project is funded by the Department of Education of Inner Mongolia Autonomous Region, Natural Science Foundation general project, NJZY22222. This project is funded by the Ordos science and technology plan project, 2022YY011.

**Data Availability Statement:** Not applicable.

**Acknowledgments:** The authors would like to thank Guixuan Wu from GTT-Technologies (Germany) for support in thermodynamics calculation with FactSage.

**Conflicts of Interest:** The authors declare no conflict of interest.

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
