# Peer review of "High-Temperature Reaction Mechanism of Molybdenum Metal in Direct Coal Liquefaction Residue"

_catalysts, doi:10.3390/catal12080926_

Round 1
Reviewer 1 Report
The article has a scientific character. The work concerns the influence of molybdenum on the coal pyrolysis process. The Authors applied correct research methods and used the appropriate measuring equipment. The content of the work is logically written. The manuscript contains 7 figures and 5 tables. Figures and tables are not properly prepared. Authors cited 21 literature sources.
General remarks
1. The novelty of the work in comparison with the state of the art or science should be indicated. Authors must extend the analysis of the state of the art. They also need to show that their work brings an aspect of novelty compared to already existing solutions, in particular with your previous articles.
2. Specify the data of the apparatus in the article in the following order: device designation, manufacturer's name, city, country.
3. In general, improve the discussion of the results obtained
4. Edit the text carefully
Detailed comments
1. Verse 4 "zhu" and "WU" is it correct?
2. Enter a space between the word and the "[" sign, it applies to the entire work 3.
Fig 3 is the description below the drawing and not above it.
4. Table 1, 2, 3, 4, 5 give the data along with the measurement uncertainty (result in the form X ± δ)
5. Figures 1 ÷ 5 should be enlarged,
I recommend making a major revision
Reviewer 2 Report
In the work the experimental studies and the interpretation of the results is not satisfactory. There is no proper evidence to proof the statements put forwarded by the authors. Such studies require online analysis of the products using FTIR or XRD to confirm the formation of different phases.
1. The authors statae that MoO3 and MoS2 powder are not stable at high temperature and there are literatures cited by the authors. It would be good to show the overall mechanism involved in the transformation and retention of this catalyst.
2. The authors state that there is in the sublimation steps “The evaporation and condensation of molybdenum trioxide are often accompanied by the formation of various polymers, such as dimer (Mo2O6), trimer (Mo3O9), and tetramer (Mo4O12).” Is MoO3 generally used as as a catalyst ? If so does it require any special conditions so that it can be retained back to its original form .
3. Is molybdenum sulfide a catalyst or a reactant to produce molybdenum trioxide ?
4. For the thermogravimetric studies to interpret the reaction mechanism of DCLR(ER)-MoO3 system, from where is DCLR(ER) taken from ? What is the weight ratios of the components? It would be good show the images of the molybdenum trioxide, molybdenum sulfide and DCLR.
5. In the High-temperature reaction mechanism of DCLR(ER)-MoS2 system studies and Figure 5 the authors state state formation of MoO3 from gas phase. However the authors in the equation 1 state that different polymers are formed.
6. The FTIR characterization of catalyst is need.
7. XRD data is needed to confirm the statement of the authors regarding the changes in the phases during pyrolysis especially for molybdenum sulfide
8. The SEM images should be separated from the TG/DSC plots. The images need to be big.
9. What about the porosity studies?
Reviewer 3 Report
Reviewer report
This paper describes results of the study on pyrolysis of direct coal liquefaction residue in the presence of molybdenum trioxide and oxygen. The author investigated thoroughly transformtions of molybdenum compounds under the high temperature reaction conditions using TG/DSG and SEM methods. They show formations of various molybdenum compounds in gaseous and solid states, which take part in pyrolysis of organic components of direct coal liquefaction residue.
This paper is very good from catalytic and inorganic point of view. However, the lack of the paper is insufficient description of the behavior of organic components of the coal. One of the main points in this study is what happens with organic molecules. The authors should provide data on analysis of organic products of the coal pyrolysis by means of element analysis, gas-chromatography-mass-spectrometry techniques. It will substantially improve the paper.
One more comment on the paper is that Conclusion section should contain some perspectives of the use of molybdenum compounds for pyrolysis of direct coal liquefaction residue. The authors must compare this molybdenum method with others in the Conclusion section.
Some minor comment is as follows: Equations 1, 2, 5, 6, it should be: “Mo”.
The paper may be accepted after the above-mentioned changes.
Round 2
Reviewer 1 Report
Thank you for considering my comments. I will recommend publishing the work.
Reviewer 2 Report
The work is presented in much better form when compared to the initial submission. The manuscript can be present in the current form
Reviewer 3 Report
The authors took into account most of the reviewers' comments and improved the paper very much. This paper may be accepted as it is.